# A Structural Validation of the Brief COPE Scale among Outpatients with Alcohol and Opioid Use Disorders

**DOI:** 10.3390/ijerph20032695

**Published:** 2023-02-02

**Authors:** Melissa Kadi, Stéphanie Bourion-Bédès, Michael Bisch, Cédric Baumann

**Affiliations:** 1UR4360 APEMAC, Health Adjustment, Measurement and Assessment, Interdisciplinary Approaches, School of Public Health, Faculty of Medicine, University of Lorraine, 54000 Nancy, France; 2Versailles Hospital, University Department of Child and Adolescent Psychiatry, 78157 Versailles-Le-Chesnay, France; 3Health Care Centre of Accompaniment and Prevention in Addictology (CSAPA), 54520 Laxou, France; 4Methodology, Data Management and Statistics Unit, University Hospital of Nancy, 54000 Nancy, France

**Keywords:** Brief COPE, coping, structure validity, substance use disorders, confirmatory factor analysis, clustering analysis on latent variables

## Abstract

Recovery from substance use disorder requires access to effective coping resources. The most widely self-reported questionnaire used to assess coping responses is the Brief COPE; however, different factorial structures were found in a variety of samples. This study aimed to examine across outpatients with substance use disorders the factor structure of the short dispositional French version of the Brief Coping Orientation to Problem Experienced (COPE) inventory. The French version of the Brief COPE was administered in a sample of 318 outpatients with alcohol or opioid substance use disorder. A clustering analysis on latent variables (CLV) followed by a confirmatory factorial analysis (CFA) was conducted to examine the factor structure of the scale. The internal consistency of the Brief COPE and its subscales were also studied. The analysis revealed a nine-factor structure with a revised 24-item version consisting of functional strategies (four items), problem-solving (four items), denial (two items), substance use (two items), social support seeking (four items), behavioral disengagement (two items), religion (two items), blame (two items), and humor (two items) that demonstrated a good fit to the data. This model explained 53% of the total variance with an overall McDonald’s omega (ω) of 0.96 for the revised scale. The present work offers a robust and valid nine-factor structure for assessing coping strategies in French outpatients with opioid or alcohol substance use disorder. This structure tends to simplify its use and interpretation of results for both clinicians and researchers.

## 1. Introduction

On 11 March 2020, the World Health Organization declared the COVID-19 epidemic a pandemic [1], and governments around the world began restricting travel and social interactions to curtail the spread of coronavirus disease. Studies on the impact of COVID-19 and health restrictions have yielded varying conclusions regarding the effect of different health measures on substance use in urban areas [2,3,4,5]. A recent study conducted on a sample of individuals with substance use disorders (SUDs) and/or behavioral addictions found a moderate impact of these measures on craving levels in a sample of Italian adults. However, other studies have reported that the pandemic has had a significant impact on vulnerable populations, such as those with mental health disorders and SUDs, including alcohol and opioid use. Early studies comparing substance use before and during restrictive measures have reported an increase in opioid overdose admissions and deaths in the United States, North America, and France [6,7,8]. In addition, recent provisional estimates from the U.S. Centers for Disease Control and Prevention (CDC) indicate that opioid overdose deaths increased to more than 75,000 in the 12 months ending June 2021 [9]. Another study of 5738 French students, for a period before and after the implementation of the lockdown, showed that there was a link between self-regulation of stress induced by the situation and risky alcohol consumption. The results indicated a higher increase in alcohol consumption among the highest stressed students [10]. Furthermore, it appears that the coping mechanisms involved in the self-regulation of stress are related to the development of substance use disorders [11].

According to Lazarus and Folkman’s model, coping refers to the cognitive, emotional, and behavioral reactions made by every individual to manage the external and internal demands created by stressful events [12]. Two types of coping are distinguished: adaptive coping, which improves the management of the negative psychological impact, and maladaptive coping, which may lead to anxiety and dependency disorders. In the same way, the form that the coping processes adopt influences the success of the resolution of a stressful situation. Maladaptive coping is rigid or socially inappropriate while adaptive coping is considered flexible and efficient [13]. As coping strategies are considered an important influence in the development, course, and treatment outcome of diverse mental disorders including substance use disorders [14], learning to cope to better manage substance use is of interest [15]. There are several standardized scales that measure coping. The most commonly used questionnaire in the literature is the Coping Orientation to Problems Experienced (COPE) inventory and its shortened version, the Brief COPE [16]. It is a widely used questionnaire for assessing coping strategies in a situational or dispositional context and the shortened format makes it easier to use in both clinical practice and clinical research. The Brief COPE is composed of 28 items that are distributed into 14 coping strategies: active coping, planning, instrumental social support seeking, emotional social support seeking, expression of feelings, behavioral disengagement, distraction, blame, positive reinterpretation, humor, denial, acceptance, religion, and substance use [17]. Despite the Brief COPE potential usefulness, a review of the factorial structure of the scale (Table 1) found a large variability in the number of dimensions (2 to 14) in a variety of clinical samples.

The study of the factor structure of a questionnaire is an important step in assessing the validity of the questionnaire. It is imperative to ensure that the selected items accurately measure the constructs. If they do not, individual scores on the hypothesized constructs may not reflect what they are intended to measure. Previous international research on the factor structure of the Brief COPE has been inconclusive (Table 1), so the extent to which it measures the theoretical constructs it is designed to measure in substance abuse patients needs to be assessed. To overcome this difficulty, the researchers use several methodological approaches such as principal component analysis and exploratory analysis, using different statistical methods such as orthogonal or oblique rotation (Table 1). They allow the aggregation of items representing a similar dimension. Methodologically, the objective of exploratory factorial analysis (EFA) is to obtain a simple structure, so that each item is associated with a single factor. Nevertheless, in practice it is possible for an item to be significantly related to two or more factors, leading to a phenomenon of factor complexity (FC). This phenomenon implies making a subjective choice in the allocation of the item, which can lead the user to erroneous conclusions about its dimensionality [18,19].

Although there are previous French studies on the validity of the Brief COPE, there is no study that has examined the factor structure of the Brief COPE in the field of addictology and none that uses the clustering method [16]. So, using a new methodological approach based on clustering analysis adapted to latent variables (CLV) followed by confirmatory factor analysis based on modification indices (CFA), this study aims to evaluate the factor structure of the Brief COPE in a French sample of outpatients with alcohol or opioid use disorders.

**Table 1 ijerph-20-02695-t001:** Characteristics of the studies in the brief review on the factorial validation of the COPE Brief.

Citation	Country	N and Sample	Type of Analysis	Factorial Structure, and Items/Subscales Removed a Priori
(Richard’s and et al., 2021) [20]	Argentina	504 elderly	EFA/CFA	02 factor model
(Tang et al., 2021) [21]	China	217 caregivers, cared for children with chronic illnesses	EFA	03 factor model
(Power et al., 2021) [22]	Canada	377 federally incarcerated inmates	PCA/CFA	08 factor model 5 items excluded
(Mackay et al, 2021) [23]	Canada	174 melanoma patients	EFA	08 factor model
(Azale et al., 2018) [24]	Ethiopia	385 women with symptoms of postpartum depression	CFA	03 factor model
(Muller et al., 2003) [25]	France	3846 French adults	CFA	14 factor model
(Radat et al., 2009) [26]	France	1534 adult migraine sufferers	PCA	06 factor model
(Doron et al., 2014) [27]	France	2771 university students	CFA	05 factor model
(Baumstarck and et al., 2017) [28]	France	398 cancer patients and their caregivers	PCA/CFA	04 factor model
(Pozzi et al., 2015) [29]	Italy	148 adults with an anxiety disorder	PCA	09 factor model
(Nunes et al., 2021) [30]	Portugal	269 at-risk parents.	CFA	14 factor model
(Alghamdi, M., 2020) [31]	Saudi Arabia	302 adults	PCA	03 factor model
(Peters et al., 2020) [32]	USA	189 pregnant women	CFA	13 factor model Exclusions of substance use items
(Cramer et al., 2020) [33]	USA	576 college students	CFA	04 factor model Self-blame and self-distraction removed
(Abdul Rahman et al., 2021) [34]	United Arab Emirates	423 female nurses	CFA/EFA	02 factor model Exclusions of 06 items (1, 4, 11, 18, 19, and 28)
(Matsumoto et al., 2020) [35]	Vietnam	1164 HIV-infected patients	CFA/EFA	06 factor model Two items removed on EFA

## 2. Materials and Methods

### 2.1. Participants

Data were collected from the French multicenter prospective SUBstance Users and Quality Of Life (SUBUSQOL) cohort study of outpatients with alcohol dependence or opioid dependence (ClinicalTrials.gov ID: NCT02894476). Patients aged over 18 years who started care and met the Diagnostic and Statistical Manual of Mental Disorders fourth edition criteria for alcohol or opioid dependence (DSM-IV) [36] were consecutively included in the SUBUSQOL study. Patients were recruited in four centers in two different regions of France specialized in the treatment of addictions by French clinicians certified in addictive pathologies and familiar with the DSM-IV. Patients were assigned to alcohol or opiate dependence groups according to their main type of dependence (alcohol or opiates) according to DSM-IV axis I.

### 2.2. Data Collection

All data were obtained at the time of inclusion. Coping strategies were assessed with a self-administered questionnaire. Sociodemographic and clinical data were collected through medical interviews.

#### 2.2.1. Coping Strategies

Coping strategies were assessed by the French version of the Brief COPE that assesses trait coping (the usual way people cope with stress in everyday life) as the good psychometric properties of this dispositional format have never been demonstrated [25]. The questionnaire includes 28 items, ranging on a four-point Likert-type scale from one (“not doing it at all”) to four (“I’ve been doing this a lot”), exploring 14 strategies: active coping, planning, use of instrumental support, positive reframing, acceptance, use of emotional support, denial, venting, self-blame, humor, religion, self-distraction, substance use, and behavioral disengagement (Appendix A). The score for each dimension is derived by averaging the scores of the items in the subscale. Items with high scores indicated a greater use of coping strategies [37].

#### 2.2.2. Sociodemographic and Clinical Data

Sociodemographic and clinical data were also collected such as age, gender, marital status, living arrangements, occupational status, educational level, type of substance dependence, and duration of illness.

### 2.3. Statistical Analysis

#### 2.3.1. Descriptive Analysis

Continuous variables were expressed as the means (standard deviation) or medians as appropriate for continuous variables, and categorical variables were expressed as numbers or percentages.

#### 2.3.2. Construct Validity

To assess the construct validity of the original 28-item, 14-factor structure, a CFA was conducted with a French sample of substance-abuse patients. To explore a better structure of the scale, preferably with fewer factors, a CLV (Clustering of Variables around Latent Variables) approach was carried out, using the ClustVarLV package available on R 4.0.3 [38]. The CLV approach adopts an exploratory analysis perspective. The fixation of the number of clusters was obtained by examining the dendrogram and the graph showing the evolution of the aggregation criterion (Figure 1).

In order to evaluate the adequacy of the original structure and the models derived from the CLV, several fit indices [39,40] were calculated: (a) χ^2^, a non-significant value indicates a good fit; (b) the Comparative Fit Index (CFI), the Tucker-Lewis Index (TLI), a value of CFI and TLI ≥0.95 indicates a good fit; (c) the Root Mean Square Error of Approximation (RMSEA) and Standardized Root Mean Square Residual (SRMR), a value ≤0.05 indicates a good fit; and the Akaike Information Criterion (AIC) which is a comparative indicator, a lower value of which is in favor of the best model [40].

#### 2.3.3. Reliability

The internal consistency was assessed using McDonald’s omega (ω) coefficient [41]. Cronbach’s alpha (α) is a more widely used measure of internal consistency, however, the alpha coefficient is not recommended for scales that are composed of two items. Furthermore, the ω coefficient was chosen because it does not assume that items have the same loadings and performs better than the α coefficient when errors covary [42,43]. Coefficients ϖ were estimated with R 3.3.2 and the Psych-package based on minres factoring and polychoric correlations.

## 3. Results

### 3.1. Characteristics of Sample

The characteristics of the 318 participants are described in Table 2. Of all participants, 78% were male and 22% were female. The mean age was 38.3 years (SD = 10.5); 47% were single, and 62% lived alone. Most of the sample (80%) reported being actively employed and (91%) (n = 288) had a high-school-level or university-level education. More than 50% had opioid dependence and 218 (45.9%) patients had alcohol dependence according to DSM-IV criteria. In addition, the mean duration of the disorder was 14.6 (SD = 10.7).

### 3.2. Factorial Analysis

#### 3.2.1. Clustering around Latent Variable

The dendrogram presented on the left side of Figure 1 suggests a nine-cluster partition. The graph showing the variation of the clustering criterion on the right of Figure 1 shows that the criterion makes a clear jump from seven to six clusters. This means that the loss of cluster homogeneity is significant with six clusters and that a partition into seven clusters should be retained. The partition into K = 7 and K = 9 clusters, available with the synthesis command (cope_clv, K = 7/K = 9), recovered the blocks of the adaptation mechanisms perfectly. The partition in nine clusters presented a better conceptual relevance, and explained 60.7% of the total variance.

#### 3.2.2. Selection of Fit Models: Confirmatory Factor Analysis

Table 3 presents the CFA results for the seven-, nine-, and fourteen-factor models. The nine-factor structure provided the best fit. The sources of misfit were examined because the CFI and TLI of the nine-factor model were low according to the recommendations of [40] (CFI and TLI ≥ 0.95 indicates a good fit). The use of the modification indices yielded a revised version of the nine-factor model including 24 items that better fit the data.

Examination of the modification indices of the nine-factor structure suggested that items 05, 09, 19, and 20 could be loaded onto other factors than those to which they were initially assigned in order to decrease the misfit. These suggestions revealed the factorial instability of items 09, 05, 08, and 17, implying a phenomenon of factorial complexity. In order not to affect scale dimensionality and interpretation, these items were removed. The modification indices also suggested covariances between four pairs of items that explicitly refer to feeling-based and work-oriented coping, the need for support/help, and self-acceptance, considering the positive aspects of the situation. Model fit of the revised 24-item, nine-factor scale showed a good fit with CFI and TLI ≥ 0.95, and RMSEA of 0.034, although other indices were borderline adequate (SRMR of 0.042) (Table 3). The internal consistency of most of these factors was high (>0.49; Table 3).

#### 3.2.3. Reliability

The 24 items that comprise the revised nine-factor scale and the univariate responses are summarized in Table 4. The item responses were combined for each of the nine corresponding subscales. The nine subscales are presented as follows: functional strategies (four items), problem-solving (four items), denial (two items), substance use (two items), social support seeking (four items), behavioral disengagement (two items), religion (two items), blame (two items), and humor (two items). Table 4 also presents the reliability of the nine subscales of the revised scale. The overall McDonald’s omega (ω) of the revised scales was 0.96, indicating good internal consistency. Most subscales of the revised scale showed good internal consistency with McDonald’s omega (ω) >0.49.

## 4. Discussion

In the present study, the original 14-factor structure was initially examined in substance-dependent patients. Our results showed that the original model was not strongly supported, implying that the validity of the domain scores calculated based on this structure is questionable. To determine the underlying structure, we performed an exploratory analysis using a clustering approach adapted to latent variables (CLV). The results of this approach argue for two models with seven and nine factors. The fit analysis of the two models showed that the nine-factor model was the best fit for the data, although with borderline CFI and TLI indices. To overcome these problems, possible reasons for the unacceptable fit of the model were explored by exploring the modification indices, and based on this, the nine-factor structure was modified due to its relatively better fit. The revised structure showed a better model fit, suggesting that some reasons for the unacceptable model fit were correctly identified.

As the nine-factor structure showed a better fit to the model, only this structure was modified based on the reasons mentioned above. As our results show, the modifications made improved the model fit for all indices. These results suggest that the revised dimensions are more appropriate to reflect coping strategies in addictology. Consideration of the modification indices indicated that items 05, 09, 19, and 20 had cross-loadings, making the structure unstable. The cross-loadings found for some items could provide conceptual justifications for relocating the item to other dimensions, and some were considered unexplainable. Here are some examples of the most significant cross-loadings. The cross-loading of self-distraction item 19 (“I did something to think about it less, such as going to the movies, watching TV, reading, daydreaming, sleeping, or shopping”) on the behavioral disengagement factor is explicable, as item 19 could also refer to a form of situational disengagement. Item 19 can be seen as potentially capturing multiple forms of coping and moving this item to another subscale could be unfruitful. When cross-loadings were inexplicable, the items were considered poor predictors of the non-target factors. For example, acceptance item 20 (“I accepted the reality of the fact that this happened”) was cross-loaded with the religion factor of the scale, providing no substantial justification for moving the item. These cross-loadings may indicate problems with scale construction or item wording, or difficulties with participants’ interpretation of items in specific contexts. Removing these items showed a good model fit for the data for all indices (X2, CFI, TLI, RMSEA, and SRMR). These results suggest that the revised dimensions are more appropriate to reflect coping strategies for addiction.

Although the problematic items removed improved the fit, it should be noted that these modifications may not work in other samples. Therefore, to precisely define the factor structure of the Brief COPE, it seems reasonable in future work to conduct replication studies of the modifications made in our study. In addition, an important question regarding the factor structure of the brief COPE is whether to account for cross-loading and to what extent this might challenge the dimensionality of the scale.

Finally, we believe that our understanding of the factor structure of the brief COPE will be further enhanced by examining in detail the reasons for this factor instability rather than admitting all the possible changes that could be made to improve the fit.

Although existing research has shown satisfactory evidence of validity and internal consistency of the Brief COPE questionnaire, the considerable number of different factor structures identified in the literature indicates a high variability of its structure. In addition to the hypotheses of problems of the formulation and/or perception of the items by different populations, some authors have considered that this instability depends somewhat on the method of analysis used [30]. Indeed, one of the major strengths of our study lies in the use of a CLV-AFC approach rather than the traditional AFE-AFC approach to assess the factor structure of the Brief COPE, which is considered more appropriate in order not to have to deal with a factory complexity situation arbitrarily. However, it is important to realize that in some cases the selection of the number of clusters may be difficult. Because, as shown in the results of our analyses (Figure 1), it can happen that the partitioning of the data is too fragmented. Nevertheless, the use of the indices of adequacy and especially the criterion of AIC and the appreciation of the relevance of the clusters constitute reliable tools for helping the decision.

With respect to internal consistency, the results were mostly good or acceptable. Our main results support the idea that the Brief COPE can be a useful tool for assessing coping strategies for addiction. The Brief COPE showed satisfactory validity and reliability properties for use in the assessment of coping strategies in patients with substance use disorders.

Our study has some limitations. First, the diagnoses were established according to the criteria of the DSM-IV, while currently the DSM-V is being used. In addition, the fact that only patients with substance dependence were included prior to the publication of the DSM-V under the condition of the DSM IV criteria limits the applicability of our findings to all the patients with SUD as the DSM-V now combines the abuse and dependence criteria into a single SUD. However, from a clinical standpoint, these semantic changes do not compromise the validity of our data and do not imply relevant bias. Then, the use of CFA may also be considered to be too stringent. As with all self-reported data, we obviously do not know the extent to which respondents were honest in answering questions. Additionally, while Exploratory Factor Analysis (EFA) relies on underlying assumptions and defines a model based on measured variables, error terms, and latent constructs, Cluster Analysis does not. Instead, it groups data into homogeneous groups based on their characteristics. Although this flexibility can be seen as an advantage, it can also be a limitation as the choice of method depends on the data set and research objective, introducing subjectivity to the process. Finally, although the reliability of our scale has been demonstrated, this study does not allow us to determine the extent to which each item measures the latent factor of interest, as item saturations were not studied, so our model has not been able to be adjusted accordingly.

This study should be considered an important but not exhaustive step in the long process of validating a measurement instrument. Other properties remain to be explored, such as criterion validity or invariance of its items across different characteristics of patients.

## 5. Conclusions

In conclusion, our findings suggest that the revised nine-factor structure of the Brief COPE demonstrated satisfactory validity and reliability properties. This study supports the use of the Brief COPE as a short, accessible, and validated measure of coping styles in substance abusers.

## Figures and Tables

**Figure 1 ijerph-20-02695-f001:**
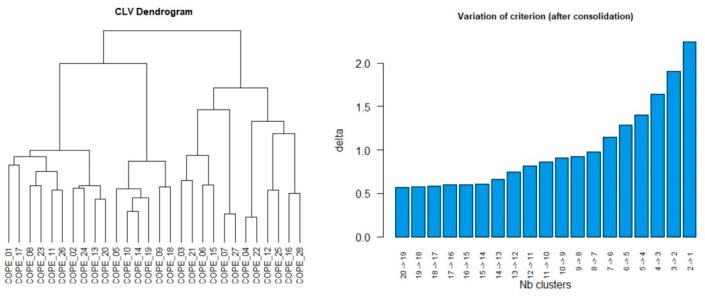
Graphs obtained by directional clustering of psychological variables. On the left, the dendrogram of the hierarchical clustering step; on the right, the variation of the clustering criterion after consolidation of the partitions using the partitioning algorithm.

**Table 2 ijerph-20-02695-t002:** Main social and clinical characteristics of the participants.

		Full Sample ^1^	Alcohol-Dependent Outpatients ^1^	Opioid-Dependent Outpatients ^1^	*p* Value ^2^
Characteristic		(n = 318)	(n = 157)	(n = 161)	
**Age**		38.4 (10.5)	42.0 (12.3)	34.7 (8)	<10^−3^ **
**Gender**					0.61
	Male	78	78	78	
	Female	22	22	22	
**Marital status**					0.002 *
	Never married	47	39	55	
	Married/living with a partner	34	34	34	
	Separated/divorced/widowed	19	27	11	
**Educational level**					<10^−3^ **
	Primary school	02	04	02	
	Middle/High school	73	59	86	
	University	25	37	12	
**Living arrangements**					<10^−3^ **
	Alone	78	46	62	
	With family	5.1	12	8.8	
	With friends	11	25	18	
	Homeless	5.9	17	11.2	
**Occupational status**					0.01 *
	Full-time work	56	64	48	
	Part-time work	24	18	30	
	Unemployed/student	17	15	20	
	Retired	03	03	02	
**Duration of addiction (years)**		14.68 (10.4)	20.42 (11.87)	10.14 (7.11)	<10^−3^ **

* *p* < 0.05. ** *p* < 0.001. ^1^ (%); Mean (SD). ^2^ Pearson’s Chi-squared test; Wilcoxon rank sum test; Fisher’s exact test.

**Table 3 ijerph-20-02695-t003:** Model fit indices of the CFA for the resulting clustering scale and the revised scale.

	Dimensions	Items	χ^2^	CFI	TLI	RMSEA	SRMR	AIC	BIC
Model 1	14	28	433.532 **	0.938	0.909	0.044	0.046	24,076.722	24,986.930
Model 2	07	28	715.074 **	0.862	0.841	0.058	0.069	24,218.264	24,515.982
Model 3	09	28	599.150 **	0.898	0.877	0.051	0.060	24,132.340	24,488.055
Model 4	09	24	300.922 **	0.962	0.950	0.034	0.042	20,931.234	21,273.211

Note: χ^2^ based on DLWS estimation; ** *p* < 0.001 CFI, Comparative Fit Index; TLI, Tucker-Lewis Index; RMSEA, Root Mean Squared Error of Approximation; SRMR, Akaike’s Information Criterion (AIC). Model 3: items 05, 09, 19, and 20 were removed from the analysis.

**Table 4 ijerph-20-02695-t004:** French revised scale reliability: omega loadings of total factors.

Items	F1	F2	F3	F4	F5	F6	F7	F8	F9
12. I’ve been trying to see it in a different light, to make it seem more positive.	1.00								
24. I’ve been learning to live with it.	0.72								
17. I’ve been looking for something good in what is happening.	0.86								
1. I’ve been turning to work or other activities to take my mind off things.	0.49								
2. I’ve been concentrating my efforts on doing something about the situation I’m in.		1.00							
25. I’ve been thinking hard about what steps to take.		0.86							
7. I’ve been taking action to try to make the situation better.		0.84							
14. I’ve been trying to come up with a strategy about what to do.		0.94							
8. I’ve been refusing to believe that it has happened.			1.00						
3. I’ve been saying to myself “this isn’t real”.			0.91						
11. I’ve been using alcohol or other drugs to help me get through it.				1.00					
4. I’ve been using alcohol or other drugs to make myself feel better.				1.00					
10. I’ve been getting help and advice from other people.					1.00				
23. I’ve been trying to get advice or help from other people about what to do.					1.00				
15. I’ve been getting comfort and understanding from someone.					0.94				
21. I’ve been expressing my negative feelings.					0.50				
16. I’ve been giving up the attempt to cope.						1.00			
6. I’ve been giving up trying to deal with it.						0.79			
22. I’ve been trying to find comfort in my religion or spiritual beliefs.							1.00		
27. I’ve been praying or meditating.							1.00		
26. I’ve been blaming myself for things that happened.								1.00	
13. I’ve been criticizing myself.								0.92	
18. I’ve been making jokes about it.									1.00
28. I’ve been making fun of the situation.									0.57

F1: functional strategies; F2: problem-solving; F3: denial; F4: substance use; F5: seeking social support; F6: behavioral disengagement; F7: religion; F8: blame; F9: humor.

## Data Availability

The data collected and analyzed during the current study are available from the corresponding author upon request.

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
