# Peer review of "A Structural Validation of the Brief COPE Scale among Outpatients with Alcohol and Opioid Use Disorders"

_ijerph, 2023, doi:10.3390/ijerph20032695_

Round 1

Reviewer 1 Report

The study "A Structural Validation of the Brief COPE Scale among Outpatients with Alcohol and Opioid Use Disorders" aimed to examine across outpatients with substance use disorders the factor structure of the short French version of the Brief Coping Orientation to Problem Experienced (COPE) inventory. The paper is well structured and clearly describes material and methods, statistical analysis, results and conclusions. 

In my opinion, the study needs only a minor revision of the language and a better composition of tables.

1. The Authors wrote:  "The score for each dimension is obtained by averaging the scores of the subscale items with higher scores indicating a higher use of the coping style". This phrase's meaning is unclear: it's possible to ease?

 Raw 53: cognitve: cognitive

Fig. 1: the legend is not visible.

Table 2 Does it miss?

Table 4 is not formatted as the Journal indicates and it's possible to improve it.

Author Response

Dear reviewer,

First of all, we would like to express our best wishes for the new year 2023. We are also deeply grateful to you for taking the time and effort to thoroughly review our work. Your constructive feedback and suggestions have been invaluable in helping us to improve the quality of this paper. We have carefully considered all the opinions expressed and have made the necessary revisions to address the points raised. We hope that these changes meet your expectations.

Sincerely,

Reviewer 2 Report

In this study, the researchers analyzed the psychometric properties of the French version of the Brief COPE Scale in order to verify its validity and internal consistency also with respect to its subscales when applied among Outpatients with Alcohol and Opioid Use Disorders.

In fact, the sample investigated was composed of 318 patients from the French multicenter prospective cohort study SUBstance Users and Quality Of Life (SUBUSQOL), whose alcohol and opioid dependence was diagnosed, however, according to criteria indicated by the DSM-IV, likely because the source Clinical Trials were dated: this is certainly a limitation since there are substantial differences from the criteria reformulated in the DSM-5, so even though reference to the previous version is specified, it would still be the case to include this condition among the limitations; nevertheless, it seems quite reasonable to assume that this condition, for the purposes of validation of this coping scale, does not involve a particularly relevant bias.

Introduction: when introducing COVID-19 in relation to alcohol and substance use some other contributions could be reported, differentiating the lock-down periods in major cities from the global effects the pandemic period: an interesting study showed the presence of a moderate psychopathological burden and low craving scores in the lockdown periods. These results are not consistent with those reported in the introduction but could give a wider perspective to the rider.

From a methodological point of view, the authors preferred Cluster Analysis on Latent Variables to Exploratory Factor Analysis on the plausible grounds that the former is limiting where the assumption, that each item is associated with a single factor, clashes with the probability that it is instead related to two or more factors; however, it must be said that while EFA defines a model including measured variables, error terms, and latent constructs, specifying underlying assumptions (a linear relationship between variables, normal distribution of them... ), Cluster analysis does not define a model, has no underlying assumptions about the distribution of the data and classify objects or cases into relatively homogeneous groups based on their characteristics, furthermore selecting the appropriate clustering method depends on the data set and the intended use for the results, so what could be considered an advantage, on the other hand can easily expose itself to the subjectivity of the researcher, whereby the data extrapolated with the statistical software are operator-dependent in departure; this should probably be clarified better in the Discussion.

This was followed by a classic Confirmatory Factor Analysis (CFA) that verified good validity for a nine-factor structure with a revised 24-item version composed of functional strategies, after verifying the fallacy of a 7-factor structure; Chronbach's alpha was used as a statistical index in this regard: there are various evidences suggesting that it rather than an index of reliability is a kind of lower limit of reliability; an alternative would have been Mcdonald's Omega.

It is noted that no reference was then made to the detection of saturations ( the relationship between the latent factor and individual items)

It would then have been interesting to apply an invariance analysis to see if the obtained model was invariant for example by gender and/or by specific disorder (opioid use/alcohol use, not in comorbidity) and generalizable.

An interesting comparison would then have been for example with other stress-related scales, such as the PSS, to study further the criterion validity of the COPE scale.

More details would be needed regarding the questionnaires, for example both to verify the completeness of the information collected. Starting again from the questionnaires also, correlations could have been made with respect to the item related to religiosity, since issues such as substance and alcohol use may be approached differently under its influence of religious belief.

From the methodological point of view, this is in any case a good piece of work, and the writing of the article appears very clear in its subparts, even the tables are well structured.

Only minor revision is recommended as far as indicated.

Author Response

(The authors gave the same response as above.)
